# An Intelligent Ball Bearing Fault Diagnosis System Using Enhanced Rotational Characteristics on Spectrogram

**DOI:** 10.3390/s24030776

**Published:** 2024-01-25

**Authors:** Gyujin Seong, Dongwan Kim

**Affiliations:** Department of Electronics Engineering, Dong-A University, Busan 49315, Republic of Korea; dlrntka7592@gmail.com

**Keywords:** filter bank, signal processing, convolutional neural network, fault diagnosis system

## Abstract

Faults in the ball bearing are a major cause of failure in rotating machinery where ball bearings are used. Therefore, there is a growing demand for ball bearing fault diagnosis to prevent failures in rotating machinery. Although studies on the fault diagnosis of bearing have been conducted using temperature measurements and sound monitoring, these methods have limitations, because they are affected by external noise. Therefore, many researchers have studied vibration monitoring for bearing fault diagnosis. Among these, mel-frequency cepstral coefficients (MFCCs) and 2D convolutional neural networks (CNNs) have attracted significant attention in vibration monitoring schemes. However, the MFCC in existing studies requires a high sampling rate and an expansive frequency band utilization. In addition, 2D CNNs are highly complex. In this study, a rotational characteristic emphasis (RCE) spectrogram process and an optimized CNN were proposed to solve these problems. The RCE spectrogram process analyzes a narrow frequency band and produces low-resolution images. The optimized CNN was designed with a shallow network structure. The experimental results showed an accuracy of 0.9974 for the proposed system. The optimized CNN model has parameters of 5.81 KB and *FLOPs* of $1.53\times 10^{6}$. We demonstrate that the proposed ball bearing fault diagnosis system can achieve high accuracy with low complexity. Thus, we propose a ball bearing fault diagnosis scheme that is applicable to a low sampling rate and changing rotation frequency.

## 1. Introduction

Ball bearings are responsible for reducing rotational friction and supporting radial and axial loads in rotating machinery. However, they are the components that most commonly develop faults in rotating machinery [1,2]. Therefore, ball bearing faults can cause severe damage to rotating machinery, resulting in economic losses and increased safety risks [3,4,5,6]. Over the past few decades, many studies have been conducted on ball bearing fault diagnosis [7,8,9,10,11,12,13,14,15,16]. Ball bearing fault diagnosis methods can be categorized into temperature measurement, sound monitoring, and vibration monitoring. Temperature measurement utilizes 2D ball bearing images that are acquired using an infrared thermal imaging camera to diagnose ball bearing faults, whereas sound monitoring utilizes ball bearing sound signals that are acquired using a microphone to diagnose the faults. However, sensors used in conventional methods have certain limitations. In the case of infrared thermal imaging cameras, dust and air particles in the field can interfere with the image information and distort readings. In the case of microphones, external noise distorts the signal, and sound attenuates over the measurement distance. However, accelerometers are robust to external factors, because they measure vibration signals when in contact with the rotating machinery. Table 1 summarizes the sensors and the limitations of the measurement methods that are used for bearing fault diagnosis.

Convolutional neural networks (CNNs) have powerful nonlinear feature extraction capabilities, so many researchers have recently applied CNNs to classify imaged bearing vibration signals [17]. Imaged bearing vibration signals include time and frequency information. In particular, the short-time Fourier transform (STFT), wavelet transform (WT), and Hilbert–Huang transform (HHT) are used to analyze vibration signals in terms of time and frequency. Among them, the STFT is a simple method for converting vibration signals into time and frequency. Therefore, the STFT is more efficient than the WT and HHT [18]. Shan et al. [19] proposed Mel-CNN, which combines an MFCC image and a CNN by utilizing sound signals. The Mel spectrogram effectively extracted bearing fault features due to the low correlation between vectors, and the CNN was combined with this for image classification to achieve effective fault classification. However, the sound signal contains not only bearing fault features but also a large amount of white noise. Furthermore, the noise in real industrial plants is critical for ball bearing fault classification. Xia et al. [20] proposed a CNN model for fault diagnosis during vibration monitoring. This method used a method that arranged the collected multichannel vibration signals in a row and preserved the time and space information of the original signals, so that the rich information effectively diagnosed the fault. However, existing CNNs are limited in their application to fault diagnosis because they require numerous training samples to achieve high diagnostic accuracy. In [21], a bearing fault diagnosis method based on Mel frequency cepstral coefficients (MFCCs) and a CNN was proposed. First, an MFCC was introduced to extract the signature features of the fault signal. Subsequently, a CNN with deformable convolutional blocks was used to extract spatial information between categories and improve the extracted spatial frequency features. This shows that the MFCC can be used as a tool for feature extraction in bearing fault diagnosis. The MFCC is effective for the feature extraction of speech signals. However, the MFCC used in existing studies requires a high sampling rate and utilizes a wide frequency band, even when using vibrating signals [22,23,24]. In addition, the CNN increases the complexity of the fault classification process [25,26,27]. Within the same context, [28] extracts the characteristic fault frequencies from the resonant frequency band of the vibration signals. The resonant frequency band is a relatively higher frequency band than the ball bearing’s fault frequency.

To overcome the limitations of existing research, we proposed a ball bearing failure diagnosis system, which consists of two steps: feature extraction and fault classification. The feature extraction method for the proposed system is a rotational characteristic emphasis (RCE) spectrogram, which modifies the MFCC process to emphasize the ball bearing’s fault features. In particular, the Mel scale was modified to improve the resolution of the fault frequency. In addition, the discrete cosine transform (DCT) was not used in the classification, because it reduces feature information and requires additional computation. Next, the fault classification of the proposed system was an optimized CNN, which uses the RCE spectrogram as input image. Furthermore, the optimized CNN was designed to fit the size of the RCE spectrogram and optimized for effective classification. The ball bearing fault diagnosis system is illustrated in Figure 1. The contributions of this study are as follows:The proposed system extracts the fault features of ball bearings that are related to rotational frequency. This method does not require a high sampling rate for the accelerometer when collecting vibration signals.Compared to existing studies that use MFCCs to extract the features of vibration signals, the RCE spectrogram process analyzes a narrow frequency band. This requires fewer resources than existing feature extraction methods. In addition, the optimized CNN architecture for fault classification has lower complexity than existing CNN architectures.The Case Western Reserve University (CWRU) dataset was used to evaluate the ball bearing fault diagnosis. The experimental results indicated that the proposed method classified ball bearing faults with an accuracy of 0.9974. This satisfies the conditions for a ball bearing fault diagnosis system.

The remainder of this paper is organized as follows: The background for understanding ball bearing fault characteristics is explained in Section 2. The proposed ball bearing fault diagnosis system is described in Section 3. In Section 4, we describe an experiment using the proposed system and analyze the experimental results. Finally, the conclusions are presented in Section 5.

## 2. Background Knowledge

Ball bearing fault frequencies occur periodically when a component of a ball bearing is faulty. The ball pass frequency of the outer race (*BPFO*) is observed when the outer race of a ball bearing is faulty. The ball passes through the fault in the outer race, causing a periodic vibration. The ball pass frequency of the inner race (*BPFI*) is observed when a fault occurs in the inner race of a ball bearing. The ball spins and passes through the fault in the inner race, causing a periodic vibration. The ball spin frequency (*BSF*) is observed when a fault occurs in the balls of a ball bearing. As the balls pass through the bearing, they spin around their own axis and generate periodic vibrations. All of these cause periodic vibrations, resulting in a peak being observed in the frequency domain. The fault frequency for different types of faults can change depending on the specifications of the ball bearing and the rotational frequency of the rotating machinery. Equations (1)–(3) show the fault frequencies according to the types of ball bearing faults:(1)$$BPFO=\frac{{nf}_{r}}{2}\left\{1+\frac{d}{D}cos\alpha \right\}$$
(2)$$BPFI=\frac{{nf}_{r}}{2}\left\{1-\frac{d}{D}cos\alpha \right\}$$
(3)$$BSF=\frac{{Df}_{r}}{2d}\left\{1-{\left(\frac{d}{D}cos\alpha \right)}^{2}\right\}$$
where D, d, a, n, and $f_{r}$ are the pitch diameter, ball diameter, contact angle between the ball and the cage, number of rolling elements, and the rotating speed of the bearing (Hz), respectively. For the ball bearing fault diagnosis dataset, we used the CWRU dataset, which contains vibration signals from ball bearings that are used when operating rotating machines [29]. The vibration signals were collected using an accelerometer attached to the rotating machine. Table 2 lists the specifications of the CWRU dataset that is used to determine the fault frequency.

The rotational frequency multiples (in Hz) in Table 3 are multiples and have no units, so Table 3 shows that the fault frequency can be detected within six times the rotational frequency. The rotational frequencies in the CWRU dataset are 29.95, 29.53, 29.17, and 28.83 Hz, with a maximum fault frequency of about 162 Hz. Bearing fault frequencies typically occur in the frequency band below 500 Hz [30,31]. This means that a sampling rate of more than 1 kHz is required for the detection of a ball bearing fault. The MFCC has been proven to be effective in extracting ball bearing fault features by existing studies. However, these existing studies require a frequency band of at least 8 kHz, based on the fact that humans can determine bearing faults by sound. Therefore, a sampling rate of at least 16 kHz is required [22,32]. We prove that ball bearing fault diagnosis is possible by effectively extracting the fault frequency features of vibration signals using frequency bands within 500 Hz.

## 3. Proposed Ball Bearing Fault Diagnosis System

### 3.1. Feature Extraction Method

The down-sampled vibration signals were used as inputs for the feature extraction method. The RCE spectrogram, which is the feature extraction method used in our proposed system, enhances the faulty features of the ball bearing and converts it into a 2D image. The RCE spectrogram process is shown in Figure 2.

The down-sampled vibration signal was filtered with a pre-emphasis to balance the frequency spectrum. If the pre-emphasis filtered vibration signal is $x_{1}[n]$, the down-sampled vibration signal is x[n], and the pre-emphasis filter coefficient is α. The pre-emphasis filter is then as follows:(4)$$x_{pre}\left[n\right]=x\left[n\right]-\alpha x\left[n-1\right]\;,\;\;\;\;\;\;n\geq 1\;,\;\;\;\;\;\;n\in \left\{1,\;2,\;3,\ldots \right\}\;$$

Next, we used the STFT to generate an image containing time and frequency information. The STFT result, pre-emphasis filtered vibration signals, and window function are x(m,w), $x_{pre}[n]$, and w[n], respectively. The STFT result is then as follows:(5)$$x(m,w)=\sum_{n}x_{pre}\left[n\right]\;w\left[n-m\right]\exp\left(-jwn\right)$$
where m is the time index, w is the frequency index, exp(−jwp) is the complex exponential function, and j is the imaginary unit. The frame length was set to 100 samples, and each frame was set to overlap by 50 samples, resulting in 19 frames. The number of overlapping samples in each frame is related to the features of the STFT. When the STFT spectrum has a large overlapping area in a frame, there is less change or distortion, but more computation. To minimize the amount of computation without losing fault features from windowing, we overlapped 50 vibration signal samples, which is half the number of frame samples. Each frame uses a fast Fourier transform (FFT) for frequency analysis. The results of the FFT are filtered using an RCE filter bank to generate a spectrogram.

The RCE filter bank is a triangular filter bank that is generated based on the RCE scale. For the RCE filter bank to detect ball bearing faults, the RCE scale was designed to have a high resolution at the fault frequency of the ball bearing. The RCE scale used to generate the RCE filter bank is expressed as follows:(6)$$R\left(f\right)=6a\cdot \frac{\exp\left[f\left(10^{-\left(c+1\right)}\right)\right]-1}{\exp\left[3a\left(10^{-\left(c+1\right)}\right)\right]-1}\cdot \frac{\exp\left[3a\left(10^{-\left(c+1\right)}\right)\right]}{\exp\left[f\left(10^{-\left(c+1\right)}\right)\right]+\exp\left[3a\left(10^{-\left(c+1\right)}\right)\right]}\;\;,\;\;\;\;\;\;\left\{0\leq f\leq 6a\right\}\;$$
where R(f) is the RCE scale, f is the input frequency, a is the rotational frequency, and c is a slope factor. The RCE scale decreased as the slope shifted from the center frequency. To control the slope, the slope factor was set to between 0 and 1. When the slope factor was zero, the RCE scale had the highest slope, and when the slope factor was one, the RCE scale approximated a first-order function. This allowed the RCE filter bank to control the resolution in the frequency band. The rotational frequency was set as a parameter in the RCE scale to allow the RCE filter bank to be used on rotational machinery at any speed.

The RCE spectrogram was generated with dimensions of 19×19×3. Therefore, an RCE filter bank with a size of 19 was required. One triangular filter has two different lines. To generate the two different lines, we need three points: a start point, a center point, and an end point. If you were to arrange these triangular filters, the start point of the next triangular filter becomes the center point of the first triangular filter, and the center point becomes the end point. Each point is an RCE scale value, so the total number of RCE scale values needed should be added by 2 to the number of triangular filters. If the number of RCE scale values is n, i takes the values 1, 2, 3, …, n. We called the set of each RCE scale value an RCE point and defined the RCE points as R[i]. The following process is required to generate the RCE filter bank. First, the rotational frequency must be six times that of the 21 equal parts to obtain the values. Each set of values is called a frequency point, which is defined as F[i]. Next, F[i] is substituted into the RCE scale to obtain the RCE points. Three consecutive values of RCE points generate one triangular filter. This process is repeated to form the RCE filter bank, which can be defined as $H_{R}(x).$ The frequency points, RCE points, and RCE filter bank are expressed as follows:(7)$$F\left[i\right]=\frac{R(6a)}{\max\left(i\right)-1}\left(i-1\right),\;\;\;\;\;\;i\geq 1,\;\;\;\;\;i\in \left\{1,\;2,\;3,\ldots ,\;n\right\}$$
(8)$$R\left[i\right]=6a\cdot \frac{\exp\left[F\left[i\right]\left(10^{-\left(c+1\right)}\right)\right]-1}{\exp\left[3a\left(10^{-\left(c+1\right)}\right)\right]-1}\cdot \frac{\exp\left[3a\left(10^{-\left(c+1\right)}\right)\right]}{\exp\left[F\left[i\right]\left(10^{-\left(c+1\right)}\right)\right]+\exp\left[3a\left(10^{-\left(c+1\right)}\right)\right]}\;\;\;$$
(9)$$H_{R}(x)=\left\{\begin{array}{c} \begin{array}{cc} \;\;\;\;\;\;\;0,\;\;\;\;\;\;\;\;\;\;\;\;\;\;\;\; & x<R\left(i\right) \end{array}\;\;\;\;\;\;\;\;\;\;\;\;\;\;\;\;\;\;\;\;\;\;\;\;\;\;\;\;\\ \begin{array}{cc} \frac{x\;-\;R(i)}{R\left(i\;+\;1\right)\;-\;R(i)},\;\; & \;\;R\left(i\right)\leq x\leq R\left(i\;+\;1\right)\;\;\;\; \end{array}\\ \begin{array}{c} \begin{array}{cc} \frac{R\left(i\;+\;2\right)\;-\;x}{R\left(i\;+\;2\right)\;-\;R(i\;+\;1)}, & R\left(i+1\right)\leq x\leq R\left(i+2\right) \end{array}\\ \begin{array}{cc} \;\;\;\;\;\;0, & \;\;\;\;\;\;\;\;\;\;\;\;\;\;\;\;\;\;x>R\left(i+2\right)\;\;\;\;\;\;\;\;\; \end{array} \end{array} \end{array}\right.$$

Figure 3 shows the RCE scale and RCE filter bank when the rotational frequency is 29.95 Hz (1797 rpm) and the slope factor is 0, 0.3, 0.6, and 0.9.

### 3.2. Fault Classification

The optimized CNN was designed to achieve accurate image classification with low computational complexity. Table 4 provides detailed information regarding the layers of the CNN model, whereas Figure 4 provides a visual representation of the optimized configuration of the CNN model.

The goal of the optimized CNN is to minimize computational costs, enhance generalization, and reduce the number of features. To achieve this goal, the optimized CNN uses small-sized RCE spectrogram images as input and consists of a cross-channel normalization layer and a global average pooling layer. The small-sized input image simplifies the convolutional operations, and the cross-channel normalization layer aggregates the information of adjacent maps and improves the generalization of the time–frequency spatial information obtained from the STFT [33]. The global average pooling layer improves the correspondence between feature maps and classes and significantly reduces the number of feature maps. Consequently, feature maps can be interpreted as confidence maps, and overfitting is mitigated, because no additional parameters require optimization [34]. Our approach involves a fully connected layer and softmax function to determine the probability distribution for classification, incorporating dropout to enhance generalization. For layers requiring an activation function, the clipped rectified linear unit (ReLU) activation function combines the advantages of the ReLU activation function with reduced computational costs and addresses issues relating to exploding activations [21,24,35].

The optimized CNN consists of an input layer, convolution layer, normalization layer, pooling layer, and fully connected layer. In the optimized CNN, blocks are not reused to achieve low complexity. Initially, the input image of the input layer is an RCE spectrogram image, so the size of the input image is 19 × 19 × 3. In the convolution layer, a 7 × 7 × 32 kernel is used, and the stride is set to 2 and the padding to 3. In the normalization layer, we use cross-channel normalization and generalize between each channel. The window channel size is 5. In the pooling layer, we use global average pooling. After that, we perform classification in the fully connected layer and obtain the classification result.

We chose a strategy for improving generalization and reducing the number of feature maps by following the structure of the existing 2D-CNN while reducing the complexity. To apply this, we used cross-channel normalization in the normalization layer and global average pooling in the pooling layer to reduce the number of feature maps.

## 4. Experiment

### 4.1. Experimental Setup and Dataset Description

The proposed diagnosis system was evaluated using the CWRU dataset as input. The dataset consists of ball bearings that are attached to the drive and fan ends and is measured using accelerometers. The classes for the ball bearing fault classification are assigned to the vibration signals in the normal state as the normal class and the vibration signals of the outer ring fault, inner ring fault, and ball fault as fault classes. The vibration signal had a sampling rate of 48 kHz and was provided as a MATLAB extension mat file. The vibration signal was down-sampled to a sampling rate of 1 kHz to analyze the frequency band within 500 Hz. The down-sampled vibration signal was collected from a random starting point of the 1000th sample, and the starting point was not repeated. This collection method is based on collecting ball bearing vibration signals from a plant using a ball bearing fault diagnosis system. Table 5 lists the acquisition conditions and classification details from the CWRU dataset.

The identification of ball bearings according to their faults helps companies that produce and repair ball bearings analyze future ball bearing faults. However, industrial plants that actually use ball bearings need to be able to diagnose quickly to prevent rotor failure when a ball bearing fails, and ball bearings need to be replaced regardless of the type of fault [36,37]. For this reason, in this paper, we categorize them into two classes: normal and fault. In addition, the data included in the fault class include the fault data of the outer race, inner race, and ball. The CWRU dataset has three fault types, inner ring fault, outer ring fault, and ball fault, and each fault type has three fault diameters. In this paper, we use all the provided fault datasets as fault classes.

The collected ball bearing vibration signal was converted into an RCE spectrogram with dimensions of 19×19×3, which was then used as input for training the optimized CNN. The total number of RCE spectrogram images was 6840, of which 4653 were used as training images, 1024 as validation images, and 1163 as test images. The parameters used to train the optimized CNN are listed in Table 6.

### 4.2. Evaluation Details

The accuracy, recall, precision, and F1 score were selected to evaluate the images of the ball bearing fault diagnosis system, and the parameters and floating-point operations (*FLOPs*) were selected to evaluate the complexity of the CNN architecture. *FLOPs* measure the total number of operations that are performed within a CNN model, representing the computational workload. The calculation of *FLOPs* was based on the layer structure of the CNN model. However, the parameters indicate the number of learnable parameters that are involved in the CNN architecture. First, the accuracy, recall, precision, and F1 scores were calculated using the following equations:(10)$$Accuracy=\frac{TP+TN}{TP+FN+FP+TN}$$
(11)$$Recall=\frac{TP}{TP+FN}$$
(12)$$Precision=\frac{TP}{TP+FP}$$
(13)$$F1\;score=\frac{2\cdot Precision\cdot Recall}{Precision+Recall}$$
where *TP* and *TN* denote the number of positive and negative instances that are correctly classified. Also, *FP* and *FN* denote the number of negative and positive instances that are not correctly classified. In detail, *TP* is the result of predicting a defective image as a fault, and *FP* is the result of predicting a fault image as normal [38]. Here, positive instances refer to the faulty state of the ball bearing and negative instances refer to the normal state of the ball bearing. To calculate the complexity, the parameters and *FLOPs* were calculated using the following equations:(14)$$Convolutional\;layer\;FLOPs=K^{2}\times C_{in}\times L_{out}^{2}$$
(15)$$Fully\;connected\;layer\;FLOPs=(2N_{in}-1)\times N_{out}$$
(16)$$Parameters=K^{2}\times C_{in}\times C_{out}+B$$

Equations (14) and (15) are used to calculate the *FLOPs*, and Equation (16) is used to calculate the number of parameters. In these equations, *K* is the kernel size, $C_{in}$ is the number of input channels, $C_{out}$ is the number of kernels, $L_{out}$ is the output image size, $N_{in}$ and $N_{out}$ are the numbers of input and output nodes, and B is the number of biases.

### 4.3. Analysis Result

To evaluate whether the image size and slope factor in the proposed model are appropriate for the CWRU dataset, we check the accuracy as a function of the slope factor and image size. Table 7 shows the accuracy and loss of the slope factor, and Table 8 shows the accuracy and loss of the image size. Through the analysis, the accuracy and loss of the ball bearing fault classification were obtained. The results indicated that the highest accuracy and loss accuracy and the loss of the test data were 0.9974 and 0.0182, respectively. Figure 5 shows the training process that were used to obtain the classification accuracy and the loss of training and test data of the proposed model, whereas Table 9 shows the accuracy and loss in the training and test. Table 10 shows the confusion matrix of the test data of the proposed model. Since the training and test data are changed from training to training, the accuracy and loss of the proposed technique shows robustness.

Next, the parameters and *FLOPs* of the optimized CNN were calculated using Equations (14)–(16). As a result of the calculation, the optimized CNN has a parameter of 5.81 KB and a FLOP of $1.53\times 10^{6}$.

### 4.4. Comparison of the Results with Other Methods

The RCE spectrogram was compared with another spectrogram, the Mel spectrogram, and the MFCC. The results demonstrated that the RCE spectrogram is effective for bearing fault diagnosis. Each image was trained with the optimized CNN and evaluated in terms of accuracy, recall, precision, and F1 score. The evaluation was conducted under identical conditions. The number of training images was 4660, and the size of each image was set to 19×19×3. Table 11, Table 12 and Table 13 show the confusion matrices of the spectrogram, Mel spectrogram, and MFCC.

Table 11, Table 12 and Table 13 show that the RCE spectrogram provides the highest classification accuracy when the image size is small. In addition, the Mel spectrogram and MFCC are unable to classify the normal class. We calculated the metric of each confusion matrix, and the results are shown in Table 14.

The CNNs were also compared in terms of the complexity and accuracy of the input images. Table 15 shows the complexities of the proposed CNN and CNNs that are used for ball bearing fault diagnosis in a related work [39].

The existing 2D CNN model has the disadvantages of high memory utilization and long computation time owing to the high number of parameters and *FLOPs*. The proposed optimized CNN is a 2D CNN model that uses small-sized input images and shallow depth layers. Therefore, the limitations of the high number of parameters and *FLOPs* of the 2D CNN model were improved.

## 5. Conclusions

The applications of noncontact bearing diagnosis methods are limited in real industrial plants owing to the influence of external noise. Therefore, this study proposes a ball bearing fault diagnosis system that combines the RCE spectrogram process and an optimized CNN using the contact measurement of vibration signals. The system down-samples the vibration signal to 1 kHz and then applies the RCE spectrogram process to generate an image with a size of 19×19×3. The generated image is used as an input to the optimized CNN to classify faults in the ball bearing. The feature extraction of the RCE spectrogram and the complexity of the CNN were evaluated. The evaluation results showed that the accuracy of the RCE spectrogram is 99.74%, which is the highest accuracy among the generated image sets, and the optimized CNN has 5.81 KB of parameters and *FLOPs* of $1.53\times 10^{6}$, despite being a 2D CNN model. In conclusion, our evaluation demonstrated that the problems of a high sampling rate, high utilization of existing wideband frequency analysis, and the high complexity of deep neural network models are solved. In particular, the RCE scale in the RCE spectrogram process can control the band of the rotational frequency and the resolution of the RCE filter bank by changing the slope factor. Therefore, the RCE filter bank can be applied even when the rotation frequency of the rotating machine changes. Furthermore, it can be used to analyze ball bearing faults in various rotating machineries.

In our future studies, we plan to design a prototype module with a ball bearing defect system. The prototype module will be used to apply the ball bearing fault system to a practical environment. Furthermore, we will compare and verify the performance of the ball bearing fault diagnosis system through experiments. Based on the experimental results, we will apply an algorithm so that the slope factor of the RCE scale can change adaptively.

## Figures and Tables

**Figure 1 sensors-24-00776-f001:**
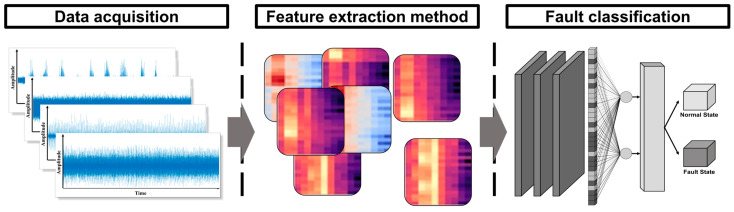
Schematic diagram of the proposed ball bearing fault diagnosis system.

**Figure 2 sensors-24-00776-f002:**
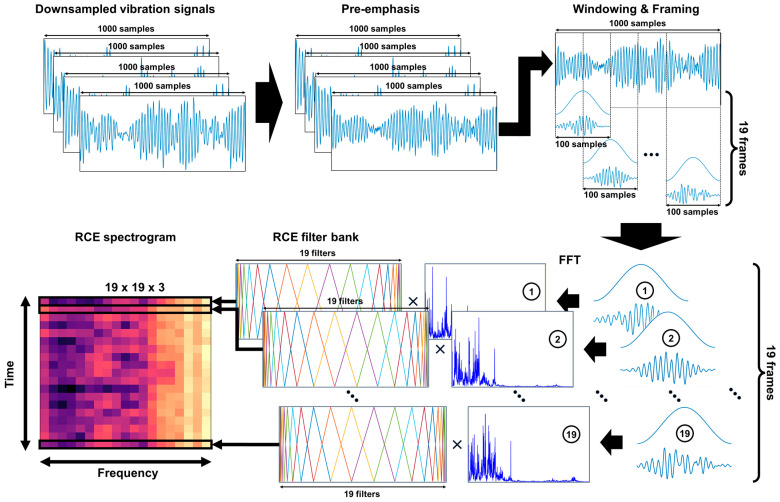
Schematic diagram of the RCE spectrogram process.

**Figure 3 sensors-24-00776-f003:**
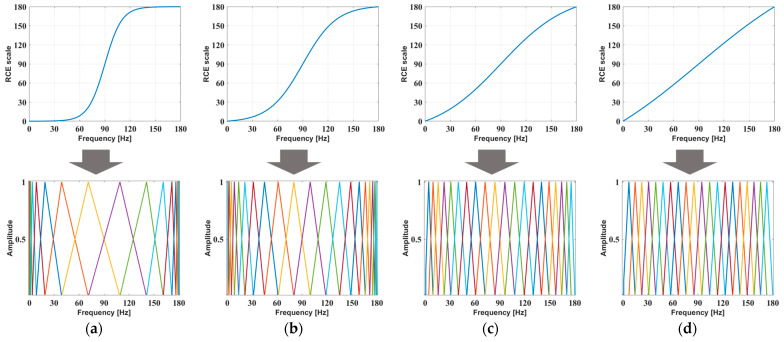
RCE scale and RCE filter bank depending on the slope factor: (**a**) c=0; (**b**) c=0.3; (**c**) c=0.6; and (**d**) c=0.9.

**Figure 4 sensors-24-00776-f004:**
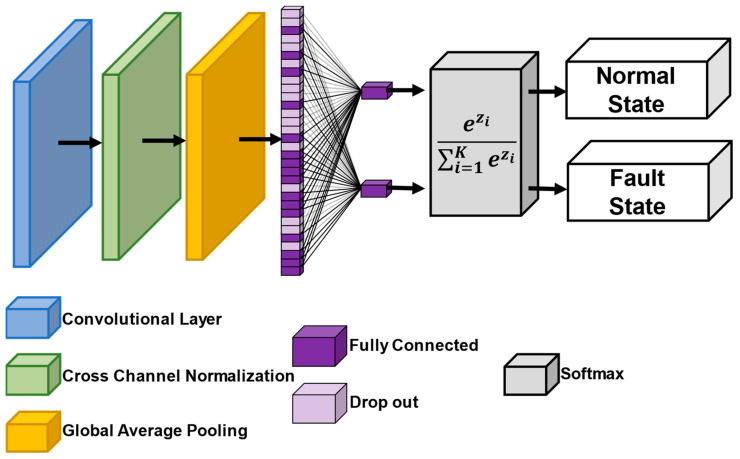
Schematic diagram of the optimized CNN.

**Figure 5 sensors-24-00776-f005:**
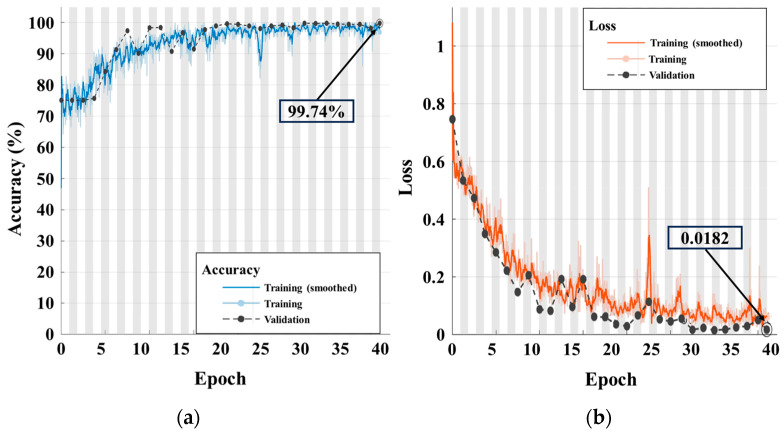
Classification training process: (**a**) accuracy; (**b**) loss.

**Table 1 sensors-24-00776-t001:** Limitations of sensor types for bearing fault diagnosis.

Method	Sensor	Limitation
Temperature measurement [7,8,9,10]	Infrared thermal imaging camera	Distortion in readings from dust and air particles
Sound monitoring [11,12,13]	Microphones	External noise and signal attenuation

**Table 2 sensors-24-00776-t002:** Specifications of the ball bearings from the CWRU dataset.

Specification	6205-2RS JEM SKF (Drive-End Bearing)	6203-2RS JEM SKF (Fan-End Bearing)
Inside diameter [mm]	25.0	17.0
Outside diameter [mm]	52.0	40.0
Pitch diameter [mm]	39.0	28.5
Ball diameter [mm]	7.94	6.75
Thickness [mm]	15.0	12.0
Number of balls	9	8
Contact angle [°]	3.134	3.126

**Table 3 sensors-24-00776-t003:** Fault frequencies according to the ball bearing fault type from the CWRU dataset.

Types of Ball Bearing Fault	6205-2RS JEM SKF (Drive-End Bearing)	6203-2RS JEM SKF (Fan-End Bearing)
	Multiple of Rotational Frequency (in Hz)	
Inner ring	5.4152	4.9469
Outer ring	3.5848	3.0530
Rolling element (ball)	4.7135	3.9874

**Table 4 sensors-24-00776-t004:** Details of the optimized CNN architecture.

Layer	Data Size	Kernel Size	Activation Function
Input data	19×19×3	Not needed	Not needed
Convolutional	10×10×32	7×7×32	Clipped rectified linear unit
Cross-channel normalization	Window channel size = 5		Not needed
Global average pooling			Not needed
Fully connected	32	32×32	Clipped rectified linear unit
Dropout	0.5		Not needed
Fully connected	2	2×32	Not needed
Output data	2	Not needed	Softmax

**Table 5 sensors-24-00776-t005:** Vibration signal acquisition conditions and classification details from the CWRU dataset.

Fault Diameter (mm)	Motor Load (HP)	Motor Speed (Hz)	Sampling Frequency (kHz)	Class
0.1778	0	29.95	48	Fault
	1	29.53		
	2	29.17		
	3	28.83		
0.3556	0	29.95	48	Fault
	1	29.53		
	2	29.17		
	3	28.83		
0.5334	0	29.95	48	Fault
	1	29.53		
	2	29.17		
	3	28.83		
0.000	0	29.95	48	Normal
	1	29.53		
	2	29.17		
	3	28.83		

**Table 6 sensors-24-00776-t006:** The optimized CNN model’s parameters.

Class	Value
Learning rate	0.01
Batch size	50
Max epoch	40

**Table 7 sensors-24-00776-t007:** The accuracy and loss of the slope factor of the training data and test data.

Slope Factor		0.1	0.2	0.3	0.4	0.5	0.6	0.7	0.8	0.9	1.0	Mean	Std.
**Training**	Accuracy	84.38%	82.81%	84.38%	93.75%	99.38%	95.31%	90.63%	82.81%	71.11%	67.36%	85.19%	9.671
	Loss	0.3774	0.4123	0.4022	0.1233	0.0455	0.1207	0.2237	0.4744	0.7582	0.7939	0.3731	0.244
**Test**	Accuracy	75.06%	68.31%	83.19%	91.13%	99.01%	90.03%	85.17%	81.21%	68.54%	64.75%	80.64%	10.70
	Loss	0.4995	0.5736	0.3980	0.1865	0.0691	0.2132	0.2626	0.5081	0.7547	0.9112	0.0691	0.252

**Table 8 sensors-24-00776-t008:** The accuracy and loss of the image size of the training data and test data.

Image Size		5× 5× 3	7× 7× 3	11× 11× 3	19× 19× 3	35× 35× 3	67× 67× 3	131× 131× 3	259× 259× 3	Mean	Std.
**Training**	Accuracy	81.81%	90.63%	92.19%	99.38%	99.22%	95.70%	98.44%	77.34%	91.96%	7.606
	Loss	0.3995	0.2237	0.1980	0.0455	0.0465	0.1202	0.0995	0.5364	0.2087	0.165
**Test**	Accuracy	75.06%	83.58%	93.64%	99.01%	99.14%	95.31%	97.59%	75.06%	89.80%	9.688
	Loss	0.5601	0.2626	0.1946	0.0691	0.0709	0.1350	0.0833	0.5616	0.2422	0.194

**Table 9 sensors-24-00776-t009:** The accuracy and loss of the training data and test data of proposed model.

Count		1	2	3	4	5	6	7	8	9	10	Mean	Std.
Training	Accuracy	100%	98.44%	99.22%	100%	99.22%	99.22%	100%	99.22%	99.22%	99.22%	99.38%	0.469
	Loss	0.0102	0.2268	0.0321	0.0124	0.0491	0.0318	0.0161	0.0161	0.0258	0.0349	0.0455	0.061
Test	Accuracy	99.74%	95.96%	99.40%	99.91%	98.19%	98.62%	99.74%	99.94%	99.40%	99.05%	99.01%	1.158
	Loss	0.0182	0.1250	0.0579	0.0545	0.0577	0.0465	0.0863	0.0863	0.0684	0.0906	0.0691	0.028

**Table 10 sensors-24-00776-t010:** Confusion matrix of the experimental results using the RCE spectrogram.

		Actual	
		Fault	Normal
Predicted	Fault	873	0
	Normal	3	287

**Table 11 sensors-24-00776-t011:** Confusion matrix of the experimental results using spectrogram.

		Actual	
		Fault	Normal
Predicted	Fault	660	286
	Normal	44	264

**Table 12 sensors-24-00776-t012:** Confusion matrix of the experimental results using Mel spectrogram.

		Actual	
		Fault	Normal
Predicted	Fault	946	0
	Normal	308	0

**Table 13 sensors-24-00776-t013:** Confusion matrix of the experimental results using MFCC.

		Actual	
		Fault	Normal
Predicted	Fault	946	0
	Normal	308	0

**Table 14 sensors-24-00776-t014:** Performance metric for each input image.

	Image	RCE Spectrogram (Proposed)	Spectrogram	Mel Spectrogram	MFCC
Metric					
Accuracy		**0.9974**	0.7368	0.7544	0.7544
Recall		**0.9966**	0.9375	0.7544	0.7544
Precision		**1.0000**	0.6977	1.0000	1.0000
F1 score		**0.9983**	0.8000	0.8600	0.8600

**Table 15 sensors-24-00776-t015:** Complexity comparison of CNNs.

Classification Types	Model	# of Parameters [KB]	*FLOPs*	Accuracy [%]
2D CNN	**Optimized CNN (Proposed)**	**5.81**	$1.53\times 10^{6}$	**99.01**
	[20]	206.32	$1.01\times 10^{7}$	99.89
	[25]	258.52	$4.69\times 10^{6}$	98.90
	[27]	367.81	$2.00\times 10^{8}$	98.77
	[26]	565.16	$8.08\times 10^{7}$	99.27
	[21]	1028.00	$4.20\times 10^{6}$	100
1D CNN	[40]	9.22	$2.6\times 10^{6}$	100
	[17]	50371.07	$1.45\times 10^{8}$	99.97

## Data Availability

Data are contained within the article.

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
