# Peer review of "An Intelligent Ball Bearing Fault Diagnosis System Using Enhanced Rotational Characteristics on Spectrogram"

_sensors, 2024, doi:10.3390/s24030776_

Round 1

Reviewer 1 Report

Comments and Suggestions for Authors

1) In table 2, what is alpha and n for both the bearing types?

2) In Table 3, what is Multiple of rotational frequency (in Hz)?

3) Below the table 3, there is description provided (Line 117 - 125), there is need for some clarity as the explanation provided is not very clear?

4) In line 164, clarify, "Thus 21 RCE scale values were required"?

5) In line 225 and 226, the authors state that "CWRU data set does not distinguish between different types of ball bearing faults and randomly collects the fault class", this not clear?

6) Give proper justification why no efforts have been made in identifying ball bearing faults

7) Is validation data, test data?

8) What was the total number of images, of which how much was training and how much was test data? 

9) Give some overview about what is TP, FP, TN and FN, in the context of this work? 

10) clearly mention how much is the training and test accuracy? How much is training and test loss? 

11) The results given in Table 8, are they for training or test data? 

12) in Line 284, the authors mention 'deep learning network'? what do they mean by it, CNN is a deep learning network!

13) With reference to table 12, focusing not only on the complexity, efforts can be made to include results pertaining to accuracy, recall, precision ad F1 score  can also be included to know the difference in the performance. 

14) The authors in order to ascertain the robustness of the model ca n validate their model using data set, which is independent of CWRU or further divide the data set in test set again into test and validation and check the performance on the validation data set. 

Comments on the Quality of English Language

The English langauge requires some revision in some places 

Reviewer 2 Report

Comments and Suggestions for Authors

This paper proposed a ball bearing fault diagnosis method based on RCE spectrogram process and an Optimized CNN with vibration signals. The CWRU experimental analysis showed that the RCE filter bank can be applied even when the rotation frequency of the rotating machine changes. The subject is interesting, however, there are several address need to be improved.

(1) In abstract, the novelty and contribution of the paper need to be enhanced.

(2) In CWRU dataset, there are several fault types, why there are just two classes (normal and fault) here?

(3) The RCE spectrogram was compared with spectrogram, Mel spectrogram, and MFCC. How about other techniques, like DWT, WPD?

(4) It is suggested to add new papers about bearing fault diagnosis, such as: 10.1109/TIM.2023.3301051, 10.1504/IJHM.2022.125092

Reviewer 3 Report

Comments and Suggestions for Authors

Your introduction and the research of previous scholars in the field is less elaborated.

In the background of Part II, please describe BPFO, BPFI, and BSF in detail.

In Figure 2, you describe a schematic of the RCE spectrogram. Are each of the 100 samples at the top a different category? Please express this clearly to avoid confusing the description.

In addition, when using the RCE spectrogram method, why was the overlapping samples chosen to be 50 and what is the rationale for this?

RCE spectrograms can give different results when using different slope factors. Does this make a difference for fault diagnosis? It is recommended to add the diagnostic results under different slope factors to the experiment.

Your proposed CNN description is rather confusing and should be described in detail the order of the stacking structure of the convolutional network. It is recommended to follow a table listing them in order.

Since you are using a wealth of data on different categories of bearing failures. In the final result, why only identify if it fails or not?

In conclusion, I don't think your experimental comparisons are sufficient.

Reviewer 4 Report

Comments and Suggestions for Authors

In this paper, a rotational characteristic emphasis (RCE) spectrogram process and an optimized CNN were proposed. The method can improve the MFCC method’s high sampling rate, expansive frequency band, and highly complex. The detailed of this method describe very clear in the theoretic parts. However, some parts need to improve so that the paper can suitable published in Sensors.

1 In the abstract, the authors declare” Approximately 44% of rotating machinery failures are caused by ball bearing faults”. Some sentences like this in the whole paper. It’s too exactly and may not accuracy. How do you think some small rotating machineries do not use ball bear?

2 The experiment part needs to stronger. Some figures can be used to show the high accuracy of this method.

3 A lot of parameters used in the new method. How to choose these parameters? Like Data size and Kernel size.

Comments on the Quality of English Language

Nothing.

Round 2

Reviewer 2 Report

Comments and Suggestions for Authors

Authors have improved the quality of the paper according to the reviewers' comments. Thus, I think it can be published in Sensors.

Reviewer 4 Report

Comments and Suggestions for Authors

The authors had down a good revision. I think it can be accepted at this situation.

Comments on the Quality of English Language

Nothing.